# Identification of Potential Drug Targets of Broad-Spectrum Inhibitors with a Michael Acceptor Moiety Using Shotgun Proteomics

**DOI:** 10.3390/v13091756

**Published:** 2021-09-02

**Authors:** Hao-Wei Chu, Bidyadhar Sethy, Pei-Wen Hsieh, Jim-Tong Horng

**Affiliations:** 1Department of Biochemistry and Molecular Biology, College of Medicine, Chang Gung University, Kweishan, Taoyuan 33302, Taiwan; sonic78123@hotmail.com.tw; 2Graduate Institute of Natural Products, College of Medicine, Chang Gung University, Taoyuan 33302, Taiwan; bidyadhar.manu@gmail.com (B.S.); pewehs@mail.cgu.edu.tw (P.-W.H.); 3School of Pharmacy, College of Pharmacy, Taipei Medical University, Taipei 11031, Taiwan; 4Research Center for Food and Cosmetic Safety, College of Human Ecology, Chang Gung University of Science and Technology, Taoyuan 33302, Taiwan; 5Department of Anesthesiology, Chang Gung Memorial Hospital, Taoyuan 33333, Taiwan; 6Research Center for Emerging Viral Infections, College of Medicine, Chang Gung University, Kweishan, Taoyuan 33302, Taiwan; 7Molecular Infectious Disease Research Center, Chang Gung Memorial Hospital, College of Medicine, Chang Gung University, Taoyuan 33333, Taiwan

**Keywords:** antiviral drug, influenza virus, itaconic acid, mass spectrometry, Michael addition

## Abstract

The Michael addition reaction is a spontaneous and quick chemical reaction that is widely applied in various fields. This reaction is performed by conjugating an addition of nucleophiles with α, β-unsaturated carbonyl compounds, resulting in the bond formation of C-N, C-S, C-O, and so on. In the development of molecular materials, the Michael addition is not only used to synthesize chemical compounds but is also involved in the mechanism of drug action. Several covalent drugs that bond via Michael addition are regarded as anticarcinogens and anti-inflammatory drugs. Although drug development is mainly focused on pharmaceutical drug discovery, target-based discovery can provide a different perspective for drug usage. However, considerable time and labor are required to define a molecular target through molecular biological experiments. In this review, we systematically examine the chemical structures of current FDA-approved antiviral drugs for potential Michael addition moieties with α, β-unsaturated carbonyl groups, which may exert an unidentified broad-spectrum inhibitory mechanism to target viral or host factors. We thus propose that profiling the targets of antiviral agents, such as Michael addition products, can be achieved by employing a high-throughput LC-MS approach to comprehensively analyze the interaction between drugs and targets, and the subsequent drug responses in the cellular environment to facilitate drug repurposing and/or identify potential adverse effects, with a particular emphasis on the pros and cons of this shotgun proteomic approach.

## 1. Michael Addition

The Michael addition reaction is defined as the 1, 4-addition of a nucleophile (the Michael donor) to an unsaturated carbonyl bond with an electron withdrawing group (EWG) (Michael acceptor) [1]. It has been widely used for the formation of chemical bonds and the production of building blocks in organic chemistry [2]. This addition reaction is a versatile method for the addition of chemical compounds containing nucleophilic groups conjugated to unsaturated compounds with electron-withdrawing substituents. The common Michael donors contain the reactive sites of thiol (-SH), amine (-NH_2_), and alcohol (-OH), and their corresponding Michael reactions are referred to as the thiol-Michael addition [3], aza-Michael addition [4], and oxa-Michael addition [5], respectively (Figure 1A).

The Michael addition reaction rate depends on the type of nucleophile and electrophile, and steric hindrance is an important element affecting the process of this reaction. For example, in organisms, peptides with side chains containing aromatic or bulky groups, such as tyrosine, threonine, and methionine, rarely or slowly react with α, β-unsaturated compounds [6]. By contrast, amino acids without steric hindrance and deactivating groups, such as phenyl or methyl motifs, can potentially undergo Michael addition (Figure 1B). Furthermore, the reaction rate of Michael addition is mainly dependent on both the ability to lose electrons and the size of the donor atoms. Thus, the reactivity of cysteine (^16^S) of the thiol-Michael donor is higher than that of lysine (primary amine, ^7^N) of the aza-Michael donor, followed by histidine (secondary amine, ^7^N) and serine (^8^O) of the oxa-Michael donor. In addition, optimizing the reaction conditions can facilitate the unfavored Michael reaction. For example, owing to the delocalized electrons of oxygen, compounds with R-OH reactive sites such as serine have poor nucleophilicity, leading to the requirement for a strong base and a higher temperature to catalyze the oxa-Michael addition (Figure 1C) [7].

Owing to its simple chemical mechanism, irreversibility, and high efficiency, Michael addition is widely applied in industries, medicine, and green engineering [8]. Michael addition has been reported as a specific mechanism for post-translational modification (PTM), and it involves covalent additions to amino acid side chains and regulates protein function in cell signaling pathways and biological processes [9]. As an electrophilic molecule, the Michael acceptor is attractive to amino acids with nucleophilic functional side chains, including cysteine [10], histidine [11], lysine [12], and serine [13]. Therefore, proteins or peptides can be modified if there are nucleophilic acids in their sequences. Because of the aforementioned reasons, we believe that the Michael addition reaction has great untapped potential in the field of drug development. Although the Michael addition has had long-term applications in the field of industrial coatings and organic reactions to yield selected compounds [14], the ability of this reaction to bring about covalent modifications in target proteins in vivo has not yet been explored. Indeed, there are several covalent drugs, such as ibrutinib [15], neratinib [16], osimertinib [17], and afatinib (Figure 2) [18], which have been used clinically for the treatment of cancer and immune diseases.

Compared with noncovalent drugs, the superiority of covalent drugs in terms of pharmacological effects is forming a stable and irreversible bond between the compound and target protein, which can block protein function when used at low doses, improve selectivity, and increase effective drug duration [19]. The study of covalent drugs can help accelerate the development of drug discovery in diseases exhibiting protein dysfunction ranging from cancer and immune disorders to infectious diseases.

In contrast to other complicated organic modifications, the basic structure of Michael acceptors, such as acrolein, can form covalent bonds with multiple target sites by a nonspecific process [20,21,22]. The specificities of Michael addition are dependent on the 3D structure and electrophilicity of β-carbon and the functional groups of Michael acceptors [23]. Although the various pairs of acceptors and nucleophiles of Michael addition provide diverse prospects in the development of organic synthesis and materials science, it results in many challenges in clinical medicine, especially in drug discovery because of high costs incurred, owing to the labor required and the complicated experiments that are performed. To overcome this problem, this review suggested a high-throughput analysis of Michael acceptor drugs and that their target proteins in complicated cell extracts should be performed using MS-based proteomics to discover suitable covalent drugs that can potentially be developed to act as broad-spectrum antiviral agents.

## 2. Recent Antiviral Drugs with Michael Addition Moiety of α, β-Unsaturated Carbonyl Group

In this review, we list the clinical antiviral drugs containing at least one α, β-unsaturated carbonyl moiety, which may ascribe an antiviral ability via the Michael addition reaction (highlighted in Figure 2, Figure 3, Figure 4 from high to low reactivity and summarized in Table 1). Although some antiviral drugs find it difficult to undergo Michael addition owing to the inappropriate regiostructure or electron transition, the others have a high probability of acting as spontaneous Michael acceptors. These compounds may have additional modes of action in addition to known functions via lower specific covalent bonding in various viral infections. We propose that the identification of these potential interactions in combination with the application of existing principles of medicinal chemistry and high-throughput proteomics opens a new avenue for antiviral drug development.

### 2.1. Retroviral Integrase Inhibitors

Retroviral integrase is used to implant viral DNA into the host genome via covalent linking during retroviruses, such as HIV infection [44]. Elvitegravir (Figure 3A), dolutegravir (Figure 4A), bictegravir (Figure 4B), and raltegravir (Figure 5A) (derivatives of aryl diketo acids) are integrase strand transfer inhibitors (INSTIs) acting as competitive inhibitors that bind to the active site Mg^2+^ of integrase by chelating hetero atoms of carbonyl and hydroxyl groups [24,25,26,45,46]. INSTIs have been used as first-line HIV drugs, not only for integrase inhibition, but also for synergistic activity with other types of antiviral drugs. Therefore, INSTIs are taken in combination with several drug types with distinct mechanisms of action, such as the combination of bictegravir with two reverse-transcriptase inhibitors, emtricitabine and tenofovir alafenamide, under the brand name Biktarvy [27]. These drugs have been reported to inhibit some protein functions to reduce virus replication, but the mechanism of covalent bonding has not been described, although all proteins contain the Michael acceptor motif. Therefore, the target peptides/amino acids are still unclear. A drug does not always affect one target, and therefore, other drug targets should be discovered through the Michael addition to uncover unknown function of drug irrespective of new applications or adverse effects. Considering the effect of only conjugated nitrogen electron pairs and/or steric hindrance on the Michael acceptor unit, the drugs elvitegravir, dolutegravir, and bictegravir might perform better Michael addition than raltegravir.

### 2.2. Influenza Neuraminidase Inhibitors

Inhibition of neuraminidase can prevent the influenza virus from escaping from its host cell and spreading the progeny virus to other cells, thus reducing viral infectivity. As structural mimics of sialic acid, both oseltamivir (Figure 3B) and zanamivir (Figure 3C) reduce viral activity by tightly binding to the active site of neuraminidase to avoid cleavage of sialic acid from the cell surface when the host is infected with influenza virus [28]. Although oseltamivir and zanamivir are highly selective and effective drugs for influenza virus, resistance to neuraminidase inhibitors was seen in a small proportion of patients infected with influenza virus, particularly during the H1N1 swine flu pandemic. There is evidence to indicate that the resistant strain retains the capacity for replication and transmission, and the mechanism of drug resistance might be compensated by the mutations in other gene fragments [47,48,49]. Interestingly, the structural motifs of oseltamivir- and zanamivir-retaining Michael acceptor moieties exhibit negligible steric hindrance and electron conjugation effects, which increases the probability of these drugs undergoing the Michael addition reaction with donor amino acids.

### 2.3. Enterovirus 3C Protease Inhibitors

Rupintrivir (Figure 3D) is an irreversible 3C and 3CL protease (3C^pro^ and 3CL^pro^) inhibitor which has been broadly applied in the treatment of picornaviruses [29] and norovirus [30]. Attacking the various features of the host’s protease enzyme helps different types of viruses to hijack cellular proteins, such as cleavage of OCT-1 in rhinovirus infection [50], cleavage of the TATA-binding proteins in poliovirus [51], induction of cell apoptosis in coxsackievirus [52], inhibition of microtubule regrowth in foot and mouth disease virus [53], and a decline in the induction of NF-κB in hepatitis A virus [54]. Notably, 3C protease inhibitors can resume these cell activities during different viral infections and act as broad-spectrum antiviral agents. In addition, rupintrivir is reported as a covalent drug that targets and forms covalent bonds with the active-site cysteine residue of 3C protease via Michael addition [55], because its α, β-unsaturated ethyl ester moiety which is easily incorporated as a Michael acceptor because it is free from other limiting structural motifs such as conjugated nitrogen lone pairs and other steric effects, thereby enhancing its ability to undergo Michael addition.

The key step in influenza virus mRNA synthesis is cap snatching from the host pre-mRNA transcripts. Baloxavir marboxil (Figure 4C) disturbs viral replication by inhibiting the cap-dependent endonuclease of the PA subunit by completing cellular mRNA and tightly binding to the active site of PA [32]. The major advantage of baloxavir marboxil in medication is the single-dose strategy, which eliminates concerns of patient adherence. However, resistance to treatment with baloxavir marboxil was reported to occur in a cohort of patients infected by influenza virus with one or two amino acid substitutions in PA [56]. Although baloxavir marboxil has two adjacent α, β-unsaturated carbonyl motifs (red bonds and atoms), the α- and β-carbon were conjugated by nitrogen lone-pair (green nitrogen), which cause carbonyl deactivation together with steric effects resulting in less or no chance of Michael addition reaction.

### 2.4. Non-Nucleoside Reverse-Transcriptase Inhibitors and Nucleoside Analogs

Distinct from nucleoside-like compounds that inhibit reverse transcriptase with their competitive mechanism, non-nucleoside reverse-transcriptase inhibitors (NNRTIs), including delavirdine (Figure 4D) and doravirine (Figure 5B), bind to the active site of viral reverse transcriptase, resulting in inhibition of HIV-1 proliferation. These NNRTIs directly block the activities of RNA-dependent and DNA-dependent DNA polymerase by interacting with the pocket site of the enzyme [33,57].

Nucleotide analogs are artificial synthetic compounds used to fight viral infection by inhibiting and destroying viral reverse transcriptase and/or polymerase with the ability to compete with natural nucleotides and stop the synthesis of viral DNA/RNA. Many nucleotide analogue drugs have been applied in clinical treatment of viral infections, such as zidovudine (Figure 4E), didanosine, and valganciclovir (Figure 5C,D) for HIV [34,35,36,58]; edoxudine, idoxuridine, trifluridine (Figure 4F–H), aciclovir, penciclovir, and valaciclovir (Figure 5E–G) for herpes simplex virus [37,38,39,40,41,59]; telbivudine (Figure 4I) and entecavir (Figure 5H) for hepatitis B virus [42]; and sofosbuvir (Figure 4J) for hepatitis C virus [43]. Mechanistic studies have shown that these drugs are processed by host kinase to achieve an active nucleotide-like form as the terminator in viral genome synthesis. All the nucleotide analogue structures conferred here have less or no probability of undergoing Michael addition covalent interactions due to steric hindrance and nitrogen electron pair conjugation.

### 2.5. C-C Chemokine Receptor Type 5 Inhibitors

C-C chemokine receptor type 5 (CCR5) is a cell membrane receptor that assists in the entry of HIV-1 into target immune cells on the surface of white blood cells [60]. Vicriviroc (Figure 5I), a CCR5 inhibitor, can prevent HIV-1 from entering the target cell by binding to the hydrophobic pocket in the extracellular site of CCR5 [61]. The specific interactions between CCR5 and vicriviroc, including strong hydrophobic interaction on Ile198 and electrostatic interaction on Glu283, were reported by Dioszegi et al. [62]. Umifenovir (Figure 5J), also known as arbidol, is commonly used for influenza treatment in Russia and China. The critical mechanism of umifenovir in antiviral processes is hydrophobic interactions with aromatic residues of amino acids, which prevents the interaction between the viral envelope and the plasma membrane of the target cell, thus leading to inhibition of viral entry into the cell [63]. A recent study showed that umifenovir can also act as a broad-spectrum antiviral drug through multiple pathways to reduce virus–cell interactions and result in antiviral activities against a variety of DNA and RNA viruses, including Zika viruses, West Nile virus, the virus that causes tick-borne encephalitis [64,65], and as a potential treatment for COVID-19 caused by SARS-CoV2 infections [66]. Tipranavir (Figure 5K) is a nonpeptidic HIV-1 protease inhibitor that forms strong hydrogen bonding interactions with the backbone active sites Asn25, Asp29, Asp30, Gly48, and Ile50 of protease to prevent cleavage of HIV gag and gag-pol polyproteins, thus inhibiting viral replication [31,67]. The chemical structures conferred here have less or no probability of Michael addition covalent interactions, due to the nitrogen electron pair conjugation and steric hindrance aspects (Figure 5I–K).

In addition to these brand/generic drugs, some compounds have been suggested as novel inhibitors during viral infection. Nuclear transport proteins, including importins, exportins (for nuclear export), and transportins are essential for regulating protein transport across the nuclear membrane. Chromosomal region maintenance 1 (CRM1) is the most important and well-known exportin involved in cell apoptosis and tumorigenesis. Thus, the most well-known selective inhibitor of nuclear export (SINE) compounds, selinexor, was developed to block the activity of CRM1 to promote cell apoptosis and inhibit tumor cell growth [68]. Apart from the treatment of cancer, several studies have indicated that some nuclear export inhibitors can recognize and inhibit the function of host CRM1, such as leptomycin B (LMB), verdinexor, and itaconic acid derivatives, for the treatment of influenza infection [69,70,71]. In particular, LMB and itaconic acid derivatives were reported to target CRM1 via Michael addition to prevent influenza virus growth by blocking viral protein transport at low drug doses. These results indicate that Michael acceptors have a high potential for the development of antiviral drugs.

Although these antiviral drugs have been widely used for virus treatment in the clinic for decades, there are still limitations in clinical therapy resulting from drug resistance, which are based on high mutation rates of viruses. Currently, most scientists discover drug-binding sites in detail by in silico molecule–molecule interaction prediction, followed by handling repetitive biological experiments. To effectively screen these Michael acceptor targeting sites, a powerful tool is urgently needed. Taking advantage of the formation of covalent bonds via Michael addition in amino acids, the proteins were treated with electrophilic molecules. Herein, we suggest that high-throughput proteomics might become a new perspective on the discovery of targeted sites of covalent drugs to investigate the mechanism of resistance to covalent drugs and to determine the feasibility of drug repurposing.

## 3. Discovery of Michael Addition Drugs by Using High-Throughput LC-MS Approach

Mass spectrometry (MS) is a high-throughput technique that benefits from enhanced high-performance liquid chromatography (HPLC) and electrospray ionization (ESI) technologies. Briefly, in HPLC, the analytes are separated from a mixture by passing them in the form of a pressurized mobile liquid phase (driven by a pump system) through a stationary phase in a column filled with solid materials [72]. Based on the characteristics of the stationary phase, columns are divided into several types, such as size exclusion, normal phase, reverse phase, ion exchange, bioaffinity, etc. The pump and the column are critical features of an HPLC separation, and they directly influence the resolution of analytes for the analytical techniques that follow. Therefore, it is important to generate an appropriate gradient in the mobile phase and select a column that is suitable for the required resolution of the analytes. For example, the most common column type used in the analysis of proteins and peptides is reverse-phase chromatography, which involves sample adsorption in a hydrophobic stationary phase and elution by decreasing the polarity of the mobile phase [73]. Generally, in MS analysis, the analytes are ionized and then separated according to their m/z value in electric or magnetic fields in a vacuum. The ESI technique is widely applied to produce ions from a liquid phase after LC by spraying the liquid under high pressure and voltage to generate an aerosol, which creates charged droplets [74]. These ions are then transported to a mass spectrometer, and the mass-to-charge ratio is measured to produce mass spectra. A mass spectrum can be generated from a single molecule and provides information regarding elemental composition, isotopic signatures, and signal intensities [75]. Considering the wealth of spectral data produced from MS, it has been suggested that these data should be annotated using database search engines, such as Mascot [76], Sequest [77], and X!Tandem [78]. This would aid in calculating the ion score for each peptide spectrum match (PSM), and spectra could be more easily matched against specific proteome databases for protein identification. Moreover, the tandem MS (MS/MS) system provides comprehensive information on analytes, such as peptide mass fingerprinting and modification. In combination with bioinformatics, the MS results were integrated to identify protein ID, PTMs, and protein quantification. Owing to their advantages for protein analysis, MS-based approaches have been used for the determination of proteome-wide PTMs in many studies [79]. Recently, Pierce et al. compared the reactivities of acrolein between cysteine, histidine, and lysine residues by analyzing acrolein-treated peptides with an ESI-tandem MS system and revealed that cysteine had the highest reactivity with acrolein [20]. On the other hand, Zhang’s group resolved a series of Michael-Michael-retro Michael addition adducts catalyzed by 9-Amino-9-deoxyepiquinine using an ESI-MS/MS system [80]. These studies revealed an emerging method to effectively analyze the chemical adducts of Michael addition.

In the discovery phase of drug development, the efficacy of drug candidates, whether from natural products or chemical synthesis, is estimated by drug tests in vitro to narrow down the drug list for further verification. After selection, these candidate compounds are indicated as functional antiviral agents to disturb the progress of the viral life cycle, including entry into host cells, replication, and release from host cells, by probably interfering with some biological processes of host cells or directly targeting foreign pathogens during infection. However, the detailed mechanism of drug action, which provides insight regarding the potential side effects and approaches to drug resistance in either cell-targeted or virus-targeted drugs, includes various perspectives, and the exploration of the mechanism of drug action in cells is not as effective as that of the drug efficacy because of the high complexity in the cellular microenvironment.

To study the mechanism of action and binding site of covalent drugs with high efficiency, we suggest that the analysis of drug-targeted proteins from cell extracts is performed using shotgun proteomics. As mentioned above, the binding sites of covalent drugs are major in cysteine, lysine, histidine, and serine in cells via Michael addition (Figure 1). With a comprehensive database and high-sensitivity mass spectrometer, the drug-targeted proteins can be identified with the proteome database and the setting of specific modification of the drug in any potential amino acid. The advanced technique and process standardization of MS-based proteomics will prove to be largely beneficial; once the platform for analysis of drug-targeted proteins is established, a routine experiment can be performed for every covalent drug by simply adjusting the setting of modification types and targeting sites. The advantage of MS-based proteomics in the identification of drug-targeted proteins is not only to indicate multiple targets of the cell line but also to analyze the proteome of both the pathogen and the host, based on their respective database in a single experiment (Figure 6).

Drug resistance is a major challenge in virus treatment, and scientists must develop new drugs for the viral strains that develop resistance repeatedly. Due to the high mutation rates of the virus, the drugs might be eliminated easily when the antiviral activity declines, resulting from a single-point mutation. Drugs that directly bind to the active site of a specific enzyme in viral particles are especially affected by mutations. Thus, identifying the drug-targeted sites on the viral proteins can help to elucidate the mechanism of drug action and the cause of antiviral resistance.

Before clinical trials for a candidate drug, the cytotoxicity and potential side effects in the human body need to be investigated as important as the antiviral effect. Unfortunately, information on the interaction in complex body envelopment is difficult to obtain using a simple-cell platform. The animal model can provide data on the effect on organisms in actual physical conditions, although there are inevitably species differences and the high cost of animals. Using proteomics, the influences in probably biological function and relationship of multiple cell/tissue types during drug treatment can be predicted in cell-based experiment with the high-throughput analysis and bioinformatics [81]. Moreover, because of the comprehensive identification of drug-targeted proteins, the available drugs, such as generic drugs, might be observed to target other virus-associated proteins and are involved in multiple signaling pathways to reduce the infection of different virus types and can be developed for new therapeutic uses and potential broad-spectrum antiviral agents (Figure 7).

## 4. Discussion

Due to the high mutation rate, infectiousness, and lethality of virulent viruses, highly potent antiviral agents are urgently required to defeat these foreign pathogens. Therefore, phenotypic drug discovery (such as antiviral assays using cell lines or experimental animals) is in a dominant position compared to target-based drug discovery. Identification of a molecular target requires a longer period to explore the mechanism of drug action in pathogens and/or hosts [82]; however, clarification of the detailed capabilities of drugs is beneficial to enhance the feasibility of drug usage and to develop drugs to fight against subsequent resistant viruses. Covalent drugs, such as Michael acceptor drugs, are able to form strongly covalent bonds at specific sites of target proteins to block bioactivities and reduce viral infection. Because of its selective efficacy for treatment, it is important to elucidate the mechanism of action of Michael acceptors in cellular bioactivity for the development of efficient drugs. With technological progress, high-throughput analysis is widely applied in various fields, especially in cancer, pathogen infection, and other health-related issues. In the past, target discovery of antiviral covalent drugs relied on molecular biological experiments that detected targeted sites by estimating the binding abilities between drugs and every potential target motif, for example, the mutant sites of the resistant virus genome and predicted drug-related sites of host proteins. To achieve high-throughput analysis of Michael acceptor drugs, MS-based proteomics is suggested to identify the modified protein sites that are labeled by specific Michael acceptors in the host cells.

Various antiviral drugs have been used clinically for several decades. Among them, covalent drugs have strong medicinal properties by directly forming covalent bonds at the target sites of virus-associated proteins. In contrast to competing drugs, which compete with viral proteins/nucleic acids to ligands, receptors, and enzymes at high intracellular concentrations of analogous compounds, covalent drugs can act at a lower dose, reducing off-target effects and cytotoxicity. The stable covalent bonding between medicinal compounds and target proteins provides a long-term duration of action, resulting in less frequent drug dosing [83].

An example indicated that the Michael acceptor drugs provided a higher efficacy in antiviral activity than reversible covalent drugs because of the target-specific binding mechanism. The potent HCV drug, telaprevir, is used to reduce HCV NS3/4A viral protease by targeting the catalytic serine (Ser139) of the enzyme; however, similar active sites are also present in several proteases in the human body, resulting in a high-dose and nonspecific usage of this drug for HCV treatment. The development of covalent irreversible drugs was in order to solve this problem through the synthesis of compounds containing an α, β-unsaturated carbonyl group, which can irreversibly and specifically bind to the noncatalytic region of HCV protease (Cys159) via Michael addition. These Michael acceptor compounds are potent inhibitors with a low IC50 value of 2–4 nM during HCV infection [84].

Recently, the most popular technology that analyzes the interaction between drugs and targets is molecular docking, which is established in silico with the applications of machine learning and artificial intelligence (AI). Based on molecular docking simulations with a structural algorithm, the drug–protein interaction can be quickly predicted by simply providing the molecular structures of both the drugs and proteins as an input to the computer. Owing to its powerful processing ability, AI drug docking plays an important role in drug development [85,86]. Although AI drug docking dominates the analysis of drug–protein interactions, it still has certain limitations; for example, the simulation results cannot reflect the actual situation in a living body with a complex environment. The drug derivatives, protein complex formation, and indirect drug reactions in cells may be latent variables and obstacles for drug docking analysis.

As a powerful technique, several studies have used MS-based proteomics to study the interaction of drugs and their targets in a simple setting, and none have been reported for antiviral-target interactions. Liu et al. employed a label-free native MS approach to analyze the multiple charges of purified mycobacterial proteins in their native states by using nondenaturing ESI [87]. This mild electrospray method can be used to identify both noncovalent and covalent protein–ligand complexes. The phytocompound altholactone against *Mycobacterium tuberculosis* (*Mtb*) was identified in the crude extract of *Polyalthia* by phenotypic screening. Subsequently, by target screening, altholactone was identified to form a protein–ligand complex with the *Mtb* protein Rv1466 out of a panel of 37 purified potential mycobacterial drug targets by native MS [80]. In this small-scale setting, an individual purified Mtb protein was examined for its interaction with the ligand altholactone, which offers a powerful retrospective confirmation of phenotypic analysis, but native MS is not suitable for high-throughput target identification. Furthermore, ligand stability and bioactivity may be a concern in a more complex organism environment. Some drugs have been reported to be metabolized to form small molecular derivatives by enzymes such as cytochrome P450 (CYPs) in microsomes [88]. Wu et al. revealed an MS-based approach for the examination of three metabolized reactive products of triclosan, an antimicrobial agent, in rat liver microsome [89]. The rat microsome model was simulated as a living body that would metabolize assimilated drugs to specific derivatives, and these metabolites were identified using the MS system to provide information on the protein-modified portion for further target screening of triclosan. The potential reactive triclosan metabolite-protein adducts were then identified in rats by a shotgun proteomic approach using highly sensitive three-dimensional liquid chromatography (size exclusion chromatography for protein separation, strong cation exchange, and reverse phase for peptide separation) coupled with an MS/MS system. Finally, 45 proteins, which are involved in carcinogenesis and endocrine pathways by enrichment analysis, were regarded as triclosan targets, and this study could explain the adverse effects mainly induced by triclosan. Additionally, AI prediction by bioinformatics is helpful and accelerates the identification of products metabolized by target compounds in living organisms. For example, a website software, BioTransformer, provides an open-access scaffold that can predict small-molecule metabolism in several conditions, such as liver tissue and gut microbial environment [90]. Furthermore, nuclear magnetic resonance (NMR) spectroscopy is well known as a powerful tool in metabonomics to identify and elucidate metabolite structures [91]. Benefitting from available NMR spectral libraries, such as the Birmingham Metabolite Library (BML-NMR) [92] and BioMagResBank (BMRB) [93], it improves the efficiency and overcomes the challenge of metabolite identification by NMR in relatively complicated samples. These techniques help to discover the drug compounds and their potential degraded fragments, which may also covalently bind to target proteins, resulting in cellular effects and resistance to foreign infection.

With whole cell/tissue proteome profiling using the MS approach, the protein IDs and modifications were obtained via database searching based on the m/z value. According to this principle, MS can be used to discover the protein/amino acid targets of an additional covalent drug. However, the drug compounds might undergo metabolism (or hydrolysis) during absorption into organisms or under conditions of extreme pH values. For example, the lactone group of the polyketide natural product, Leptomycin B (LMB), a Michael acceptor of the karyopherin protein chromosomal region maintenance 1 (CRM1), was hydrolyzed at pH 10 [94]. Otherwise, the results of cocrystallography showed that lactone hydrolysis possibly occurred after Michael addition of CRM1, and CRM1 can stabilize the hydrolyzed form of LMB to enhance LMB–CRM1 interactions [94]. These potential drug metabolisms, such as oxidation, reduction, and hydrolysis before or after Michael addition, lead to mass changes and influence the further analysis of the MS approach. To solve this problem, any possible metabolite of the target drug should be clearly interpreted for drug-modification identification.

## 5. Conclusions

Although target-based drug discovery is not as intuitive as phenotypic drug discovery in defense against viral infections, it can offer important information containing not only the other potential drug usage but also the accompanying unforeseen effects. In addition to the direct interaction of drugs and their targeted proteins, the following signaling pathways are regulated, resulting in the alteration of several bioactivities. For example, cellular proteins might be modified by phosphorylation, methylation, acetylation, ubiquitination, and other PTMs during drug treatment. Not only the modification of drug-targeted proteins but also the actual drug responses can be detected using MS-based proteomics based on its highly sensitive and multiple-target based screening. Theoretically, although covalent interactions can be detected by the MS approach, there are still several latent problems that need to be solved. For example, drug-binding sites/peptides may be undetectable in MS when proteins are present in trace amounts or the tryptic digestion is interfered by bulky drug modification, resulting in a false-negative identification. This limitation can be overcome by optimizing the sample preparation procedure, such as protein enrichment with subcellular fractionation. Furthermore, the potential drug metabolism in organisms may cause a mass bias of covalent modification to its target protein in the MS process, such as the hydrolysis of LMB when targeting CRM1. Nevertheless, MS-based proteomic studies are a powerful tool, and they reveal the practical applications of drug-target screening with a high-throughput LC-MS system to rapidly characterize drug-targeted sites, thereby helping to interpret the mechanism of drug action and improve the efficiency of drug development.

## Figures and Tables

**Figure 1 viruses-13-01756-f001:**
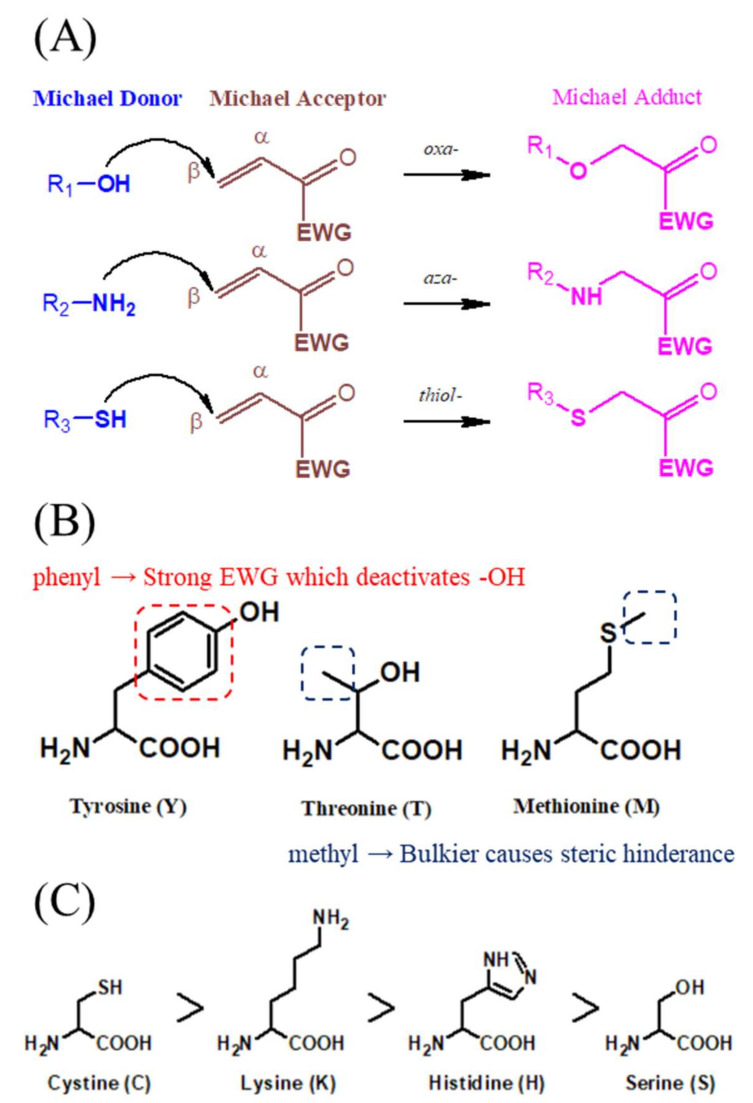
Michael addition reaction with amino acids in living organism. (**A**) Michael addition is a chemical reaction wherein a nucleophile attacks the β-carbon of an α, β-unsaturated carbonyl compound. In living organisms, the reaction is mainly classified into oxa-, aza-, and thiol-Michael additions based on the reactive site of the nucleophile being a hydroxyl, amine, and thiol group, respectively. (**B**) Amino acids are not considered to be Michael donors because of functional groups that cause steric hindrance (blue dotted line) and deactivating groups (red dotted line) (**C**) The order of reactivity preference of known amino acids as Michael donors.

**Figure 2 viruses-13-01756-f002:**
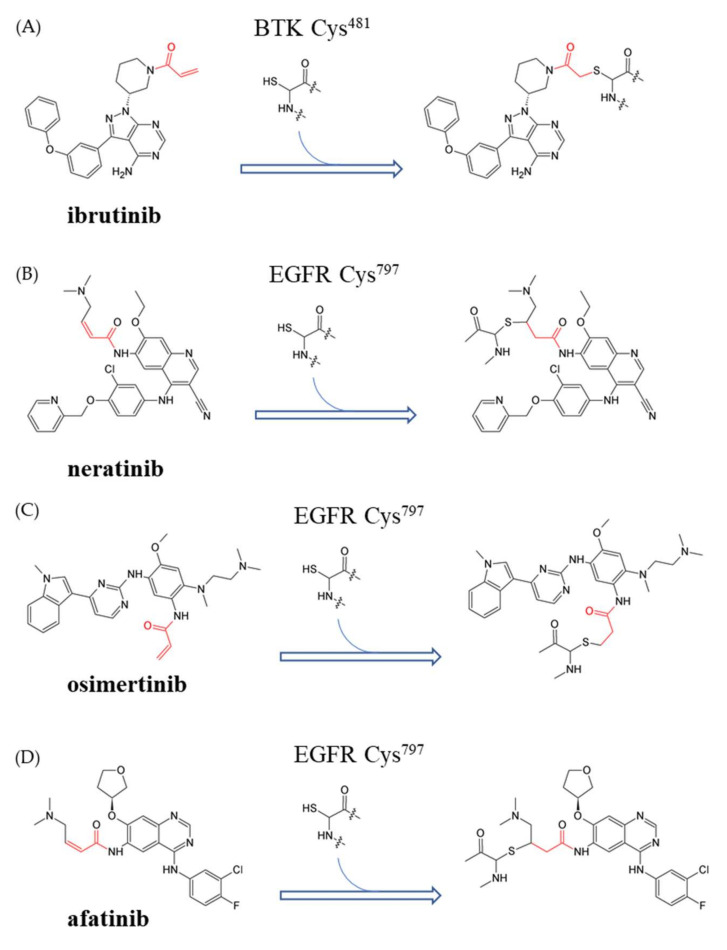
Covalent drugs bind to their target protein by forming a covalent bond with the amino acid. Ibrutinib is a covalent inhibitor of Bruton’s tyrosine kinase (BTK), and the mechanism is used to block the active site by forming a covalent bond between the α, β-unsaturated carbonyl motif, and cysteine-481 of BTK (**A**). Neratinib (**B**), osimertinib (**C**), and afatinib (**D**) have been developed for the inhibition of EGFR activity. The three compounds selectively bind to the active site (containing cysteine-797) of EGFR via a covalent bond.

**Figure 3 viruses-13-01756-f003:**
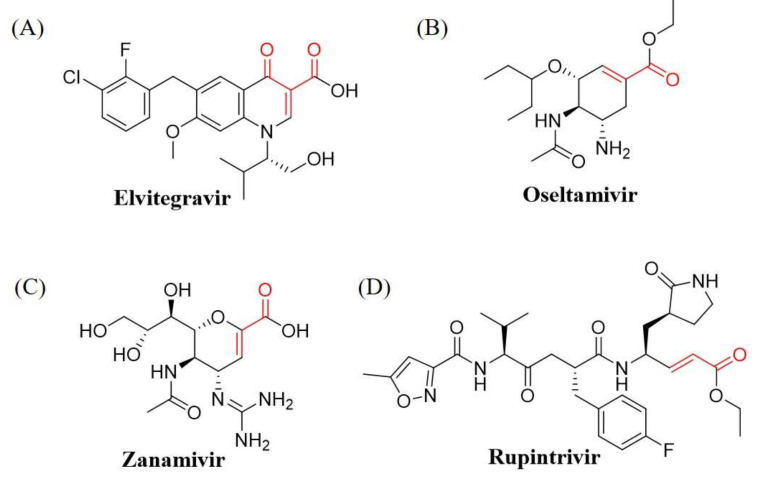
Antiviral drugs have a high potential to participate in a Michael addition reaction. The drug compounds (**A**) elvitegravir, (**B**) oseltamivir, (**C**) zanamivir, and (**D**) rupintrivir, which have a favorable Michael acceptor moiety (red bonds and atoms), have high potential to partake in a Michael addition reaction.

**Figure 4 viruses-13-01756-f004:**
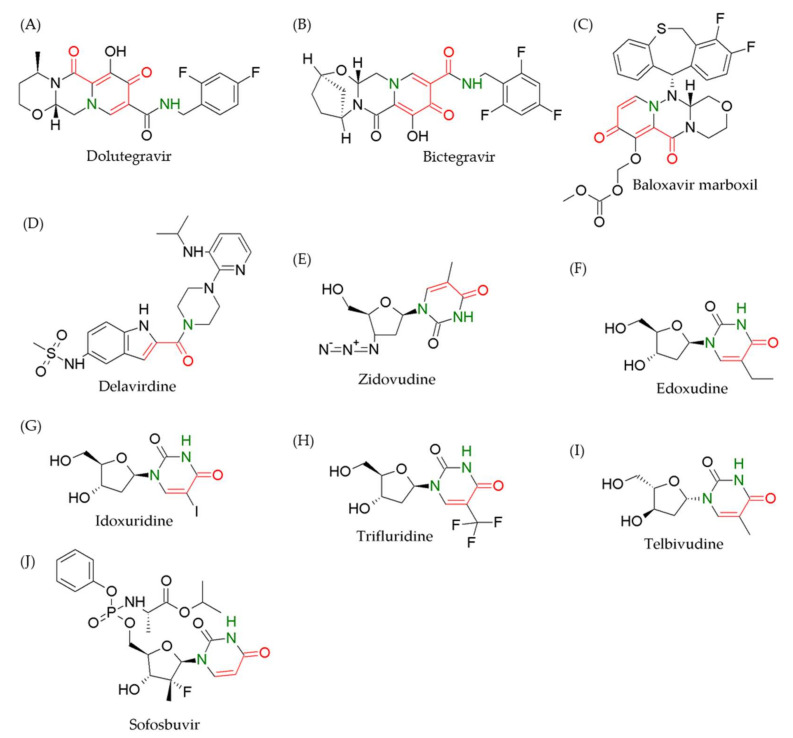
Antiviral drugs with nitrogen lone-pair conjugation near the Michael acceptor moiety. The α, β-unsaturated carbonyl motifs (red bonds and atoms) are interfered by nitrogen lone-pair conjugation (green nitrogen) to form deactivating carbonyl groups, resulting in less or no chance of the Michael addition reaction. The structures of (**A**) dolutegravir, (**B**) Biktarvy, (**C**) baloxavir marboxil, (**D**) delavirdine, (**E**) zidovudine, (**F**) edoxudine, (**G**) idoxuridine, (**H**) trifluridine, (**I**) telbivudine, and (**J**) sofosbuvir show one or two nitrogen lone-pairs causing resonance near the Michael acceptor moiety.

**Figure 5 viruses-13-01756-f005:**
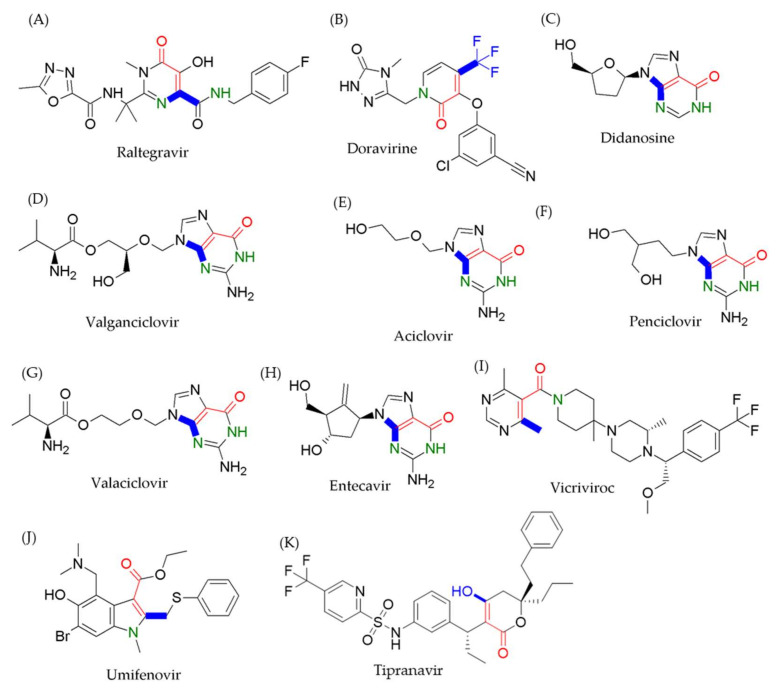
Both steric hindrance and nitrogen lone-pair conjugation disrupt the Michael addition acceptors. The structures of (**A**) raltegravir, (**B**) doravirine, (**C**) didanosine, (**D**) valganciclovir, (**E**) aciclovir, (**F**) penciclovir, (**G**) valaciclovir, (**H**) entecavir, (**I**) vicriviroc, and (**J**) umifenovir rarely undergo Michael addition because of steric hindrance (blue bond) and nitrogen lone-pair (green nitrogen) in the vicinity of the Michaela acceptor moiety (red bonds and atoms), and (**K**) tipranavir does not undergo Michael addition reaction solely because of the steric hindrance (blue bond).

**Figure 6 viruses-13-01756-f006:**
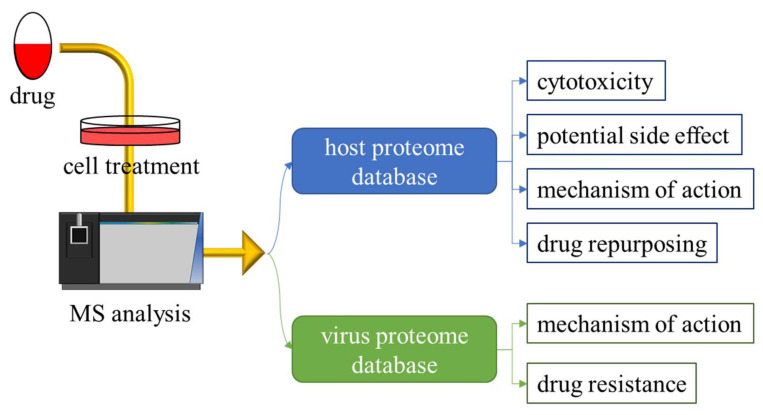
Identify drug-targeted proteins from both host and virus proteome databases. In high-throughput proteomic analysis, a large amount of information was produced in a single experiment. The results of the modification site and pathway induction/reduction were integrated from the MS analysis when the cells were treated with drugs with covalent or noncovalent interactions with drug-targeted proteins. After treatment with covalent drugs, information on drug-related proteins was analyzed with MS-based proteomics against both host and virus proteome databases to determine the mechanism of action of the drug and its potential limitations such as cytotoxicity or other side effects.

**Figure 7 viruses-13-01756-f007:**
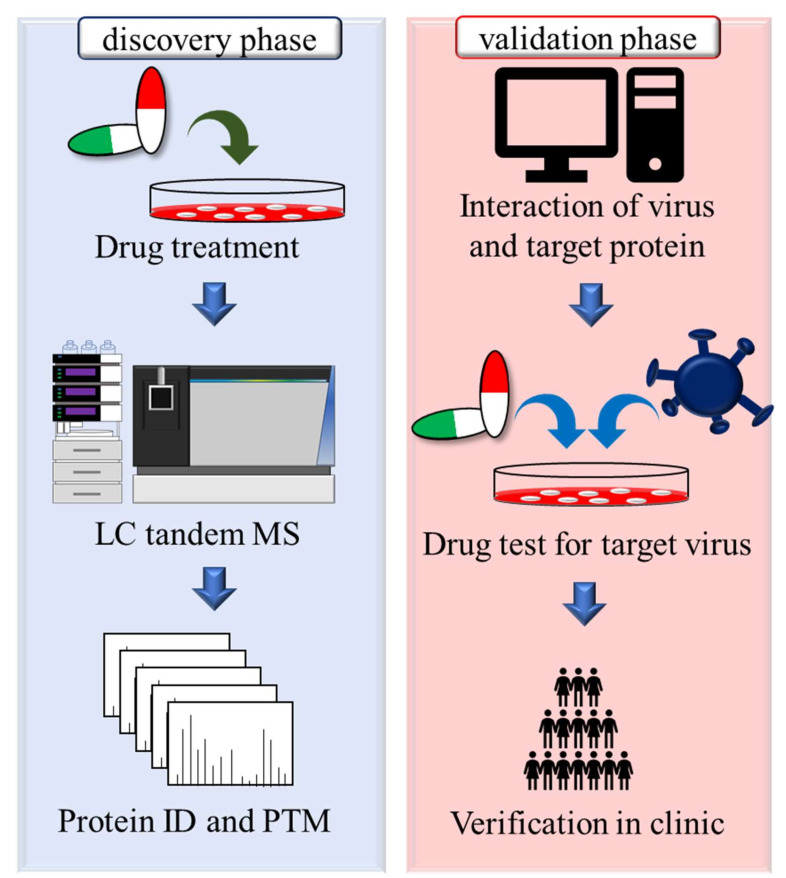
Improving the feasibility of drug repurposing with proteomics. The target proteins of a specific drug were identified during the discovery phase (**left**). The two groups of suitable and stable cell lines were treated with/without the reported drug, and these cellular proteins were analyzed using shotgun proteomics to identify the IDs and post-translational modifications (PTMs) of target proteins. These drug-targeted proteins might be involved in the viral life cycle in different viral infections. After a search of the literature, the potential viral inhibitory activity of target drugs, which are selected based on the relationship of virus and proteins, was evaluated by drug testing and further clinical trials with a large sample size (**right**).

**Table 1 viruses-13-01756-t001:** List of antiviral drugs containing an α, β-unsaturated carbonyl group.

Pharmacological Property	Drug	Target ^1^	Reference
Retroviral integrase inhibitors	Elvitegravir	HIV	Lampiris [24]
	Dolutegravir	HIV	Shah et al. [25]
	Raltegravir	HIV	Cocohoba et al. [26]
	Bictegravir	HIV	Markham [27]
Neuraminidase inhibitors	Oseltamivir	IAV	Shobugawa et al. [28]
	Zanamivir	IAV	Shobugawa et al. [28]
Protease inhibitors	Rupintrivir Tipranavir	picornaviruses	De Palma et al. [29]
		norovirus	Rocha-Pereira et al. [30]
		HIV	Vergani et al. [31]
Cap snatching inhibitors	Baloxavir marboxil	IAV	Baker et al. [32]
NNRTIs ^2^			
	Delavirdine	HIV	Chaput et al. [33]
Nucleoside analogs	Zidovudine	HIV	Mandelbrot et al. [34]
	Didanosine	HIV	Montaner et al. [35]
	Valganciclovir	HIV	Brown et al. [36]
	Edoxudine	HSV	Hamuy and Berman [37]
	Idoxuridine	HSV	Smolin et al. [38]
	Penciclovir	HSV	Spruance et al. [39]
	Aciclovir	HSV	Nasisse et al. [40]
	Trifluridine	HSV	Nasisse et al. [40]
	Valaciclovir	HSV	Beutner et al. [41]
	Telbivudine	HBV	Tsai et al. [42]
	Entecavir	HBV	Tsai et al. [42]
	Sofosbuvir	HCV	Afdhal et al. [43]

^1^ HIV: human immunodeficiency virus, IAV: influenza A virus, HSV: herpes simplex virus, HBV: hepatitis B virus, HCV: hepatitis C virus. ^2^ NNRTI: non-nucleoside reverse transcriptase inhibitor.

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
