# Peer review of "Identification of Potential Drug Targets of Broad-Spectrum Inhibitors with a Michael Acceptor Moiety Using Shotgun Proteomics"

_viruses, 2021, doi:10.3390/v13091756_

Round 1

Reviewer 1 Report

AS attached file.

Author Response

Please refer to the attached response letter.

Reviewer 2 Report

The review describes and identifies Michael addition as an important tool for covalent modification of  enzymes/ pathway proteins for target identification. This is an area of great interest and proper application of the described technology will be an attractive tool. 

However there are two major deficiencies in terms of discussion points which is recommended for the authors to include in any potentially revised submission.

 1. Covalent modification , adduct formation can be now done as a simple kit based format termed as 'controlled' protein cross linking kit. This allows crosslinking of a protein of interest in a 'pool' of proteins to which it either interact or gets within a certain proximity.  The authors should discuss any potential modification of this method that can be used for crosslinking small molecule antiviral drugs described.

They should try to describe, if the technique exist, the chemical cross linking of small molecules that do not have a alpha, beta unsaturated carbonyl moiety. 

2. The section 3 of the review which discuss the use of LC MS for identification of the covalently modified targets has a very unclear description. It seems to fall-off at the main identification part  for which the experiments will be carried. While proteomic sequencing has great throughput the process can be simplified by being able to pull down the adduct. The authors are requested to provide any insights that may simplify the target identification process. Can the target protein be isolated, should there be other simpler and lesser intense methods than advanced MS/MS .

This will allow more laboratories to use this techniques that may even allow to identify pathway proteins for understanding viral pathogenesis etc.

Round 2

Reviewer 2 Report

The authors have adequately addressed my concerns raised in the initial round of review. However I still request them to provide more insight into the methods of the LC MS target identification part of the entire workflow. 

 Please provide examples of  compatable ionization techniques ESI ??/ deconvolution methods, software used etc. and other details that will allow to nail down and identify the target protein.

Without a rigorous description of this final part the review loses the main purpose that it tries to achieve.
